# The Exploration of Flowering Mechanisms in Cherry Plants

**DOI:** 10.3390/plants12233980

**Published:** 2023-11-27

**Authors:** Yanxia Xu, Jingjing Li, Pengyi Wang, Wenhui Wang, Yuning Guo, Xueying Hao, Liyan Du, Chunling Zhou

**Affiliations:** College of Landscape Architecture and Forestry, Qingdao Agricultural University, Qingdao 266000, China; 20212110027@stu.qau.edu.cn (Y.X.); 15192496187@163.com (J.L.); 20222110035@stu.qau.edu.cn (P.W.); 20222110005@stu.qau.edu.cn (W.W.); 20212110007@stu.qau.edu.cn (Y.G.); 20212210002@stu.qau.edu.cn (X.H.); 19526513370@163.com (L.D.)

**Keywords:** cherry blossom, cold tolerance, hormone, nutrition, flower bud differentiation, secondary flowering

## Abstract

Flowering cherry (*Cerasus* sp.) are significant spring-blooming trees. However, the short blooming period and the rarity of early and late-flowering varieties limit their use in gardens in northern China. The experiment incorporated annually early-flowering species such as *Cerasus discoidea*, *Cerasus pseudocerasus* ‘Introtsa’, *Cerasus dielsiana*, *Cerasus campanulata* ‘Youkou’, *Cerasus yedoensis* ‘Somei-Yoshino’, and *Cerasus spachiana f. ascendens*, as well as twice-a-year flowering species like *Cerasus subhirtella* ‘Autumnalis’ and *Cerasus subhirtella* ‘Accolade’. We observed the timing of natural events and growth measurements for specific plants over a span of two years. This research involved a thorough examination of their ability to withstand cold temperatures, considering their physiological aspects. We examined the levels of nutrients and hormones in the flower buds at various stages of development in plants that bloom yearly and every two years. The findings indicated that *C. subhirtella* ‘Autumnalis’ is adaptable, offering the lengthiest autumn blooming phase lasting 54 days. The hierarchy of cold tolerance was as follows: *C. pseudocerasus* ‘Introtsa’ > *C. discoidea* > *Cerasus* × *subhirtella* ‘Autumnalis’ > *C. dielsiana* > *C.* ‘Youkou’. Furthermore, the soluble protein content in leaves increased before autumn flower buds’ sprout of twice-a-year flowering varieties but declined in *C. yedoensis* ‘Somei-Yoshino’ within the same time. We determined that changes in nutrient content significantly contribute to the autumn opening of *C. subhirtella* ‘Autumnalis’ robust short branch flower buds. During the final phase of flower bud development, the rise in trans-Zeatin-riboside (ZR) and indolacetic acid (IAA) promotes the initiation of the first flowering period in *C. subhirtella* ‘Autumnalis’ prior to its mandatory hibernation. The occurrence of secondary flowering involves a multifaceted regulatory process. These findings serve as valuable references for delving deeper into the mechanisms governing cherry blossom formation and secondary flowering.

## 1. Introduction

Cherry blossoms (*Cerasus* sp.) belong to the genus Prunoideae of the Rosaceae family, typically being small tree species, with about 150 variants worldwide [1]. This plant, known for its delightful white blossoms, emerges in late March during the sunny spring season, resembling bright clouds. Many cherry-blossom gardens in North China have been built, among which *Cerasus yedoensis* ‘Somei-Yoshino’ and *C. serrulata* var. *lannesiana* are the most popular varieties, but both cherry species mainly in April. Breeding early flowering and stable multi-season flowering cultivars and developing corresponding cultivation techniques are two ways to solve the problem of short and concentrated cherry flowering.

Throughout the lengthy breeding process, a cherry variety has emerged that can bloom in spring and autumn after the same year’s differentiation. This occurrence is referred to as secondary flowering. Secondary flowering can be stable (as seen in cherry blossom, peony (*Paeonia* × *suffruticosa*), and lilac (*Syzygium aromaticum*)), or unstable (as in purple leaf plum (*Prunus cerasifera*
*f. atropurpurea*), rose (*Rosa chinensis* Jacq), plum (*Armeniaca mume*), and apple (*Maluspumila* Mill)). The processes governing secondary flowering vary notably between species [2]. Ongoing studies propose that secondary flowering in plants with less predictable inheritance is mainly affected by factors like climate shifts, early leaf shedding, nutrient availability, and genetic factors. Plants carrying genetic traits for secondary flowering can display these characteristics under specific circumstances [3]. The precise mechanism of secondary flowering in perennial woody plants remains uncertain, and research on preventing secondary flowering in fruit trees has been the primary focus. For ornamental plants, however, it is desirable to encourage secondary flowering for a longer display season.

Various factors, including temperature, hormones, and nutrition, impact woody plants’ flowering, each affecting plant growth and development to varying extents. Temperature plays a vital role in constraining the growth and development of plants. Exposure to low temperatures can harm plant structures, prompt stress responses, activate the expression of relevant genes, alter the composition and structure of cell membranes, and lead to the accumulation of osmotic regulatory substances to counteract the physical and physiological harm caused by low temperatures [4,5,6,7]. Existing studies on plant cold tolerance primarily focus on temperature gradient differences and the duration of low temperatures. In the study of cold tolerance in Prunus species, scholars such as Chen [8], Zhang et al. [9], and Li et al. [10] have investigated the cold tolerance of *Cerasus* species, including *Prunus* spp. and *Prunus avium* (L.). These studies utilized artificially set temperature gradients to examine low-temperature stress’s impact on specific *Prunus campanulata* blossom’s biochemical indicators in distinct environments [11]. Researchers, such as HM Mathers [12], have studied the germplasm resources and cold tolerance of cherries. Additionally, AAG Soloklui [13] researched the cold tolerance of commercial pomegranates. Through these studies, it is evident that physiological indicators like relative conductivity, Malondialdehyde (MDA) content, Peroxidase (POD) and Superoxide dismutase (SOD) enzyme activity, and osmoregulation substance content can quantitatively signify the degree of cold tolerance among different species and varieties.

Certainly, the levels of nutrients in flower buds are significant factors that influence the flowering process of woody plants. Flower bud differentiation is a complex physiological, biochemical, and morphogenetic process [14,15,16]. When plants reach the stage of flower bud development, carbohydrates become the fundamental building blocks for transforming nutritional growth into reproductive growth. The accumulation of carbohydrates encourages the differentiation of flower buds [17,18]. Soluble sugars and soluble proteins in their leaves are continuously transported to the flower buds, providing the necessary substances and energy for bud differentiation [19]. The distribution pattern of carbohydrates, proteins, and other nutrients is closely linked to flowering as well [20]. Soluble proteins are essential structural elements in the flowering process of plants, and their levels differ significantly among various plant organs [21]. Since a substantial amount of structural substances is required for both the differentiation and flowering of flower buds, the soluble protein levels in Magnolia *soulangeana* ‘Soul-Bod’ stems decrease during the spring and summer flowering periods [20].

Plant hormones and the dynamic balance are crucial in regulating flower bud differentiation and the flowering process. The dynamic changed in abscisic acid (ABA), zeatin riboside (ZR), and indolacetic acid (IAA) contents in buds suggest an important role for these hormones during floral bud differentiation in sweet cherry [22]. High levels of ABA and IAA resulted in faster flower bud differentiation of *C. subhirtella* ‘Autumnalis’ in the early and flower primordium differentiation stages than that of *C. yedoensis* ‘Somei-Yoshino’ [23]. Higher concentrations of ABA are essential for blueberries to form multiple blossoms throughout the year [24]. External hormones can influence the awakening of flower bud dormancy. Treatment with ABA and IAA hinders the release of flower bud dormancy, whereas treatment with 6-BA, GA3, and GA4 promotes the release of dormancy [25,26]. A higher ratio of GAs to ABA was observed in sweet cherry ‘Summit’ branches from the time dormancy was broken until full bloom [27]. Exogenous GA3 treatment was the most effective way to promote the bursting of tree peony cultivars secondly [28].

Breeding early-flowering and twice-a-year-flowering cultivars is an effective way to solve the problem of short and concentrated cherry flowering. This study systematically investigated and analyzed seven early flowering and two twice-a-year-flowering cherry blossom species and varieties introduced from Wuxi, Jiangsu Province, to Qingdao, Shandong Province, in March 2016. The investigation included their phenology, growth, cold tolerance, and bud differentiation, focusing specifically on the twice-a-year flowering variety *C. subhirtella* ‘Autumnalis’ and the annual flowering variety *C. yedoensis* ‘Somei-Yoshino’. The assessment of their cold tolerance provided valuable reference for future variety introductions. The study on flower bud differentiation of secondary flowering varieties provides a preliminary understanding of the mechanism of secondary flowering.

## 2. Results and Discussion

### 2.1. Comparison of Growth Index and Phenological Period

The growth status of the five species and varieties was good, and they all exhibited normal growth states. However, their growth rates differed. As shown in Table 1, *C. dielsiana*, *C. pseudocerasus* ‘Introtsa’, and *C. subhirtella* ‘Autumnalis’ displayed slow growth from 2016 to 2017, followed by a significant increase in annual diameter increment from 2017 to 2018. The respective ranges of growth index variation were 0.66, 1.08, and 0.82, representing a significant increase compared to the previous year’s growth rate. *C. discoidea* and *C. ampanulate* ‘Youkou’ also exhibited slow growth from 2016 to 2017, with a slight increase in annual diameter increment from 2017 to 2018. Nonetheless, the increase was not substantial, with respective variation ranges of 0.46 and 0.44. The growth rate of *C. discoidea*, *C. pseudocerasus* ‘Introtsa’, and *C. subhirtella* ‘Autumnalis’ increased more slowly than other species and varieties.

In terms of the phenological period, there was not much difference between 2016 and 2017 for *C. discoidea*, *C. pseudocerasus* ‘Introtsa’, *C. subhirtella* ‘Autumnalis’, *C. dielsiana*, *C. campanulate* ‘Youkou’, and *C. yedoensis* ‘Somei-Yoshino’. The flowering period in 2017 was 2 to 3 days earlier than in 2016 (Table 2). The first flowering period of *C. subhirtella* ‘Autumnalis’ typically begins around March 15, which is 2 to 3 days earlier than *C. pseudocerasus* ‘Introtsa’, and it lasts for approximately 30 days. The second flowering period begins around September 20 and continues until early November, displaying a scattered and prolonged flowering period that can span 50 days. *C. yedoensis* ‘Somei-Yoshino’ starts its initial flowering period in early April, featuring a concentrated flowering period lasting approximately 20 days. All six species and varieties share the trait of flowering before the leaves emerge, transitioning to the leaf development phase when the entire group reaches the final flowering stage. The five introduced species and varieties enter the leaf development phase in mid-early April, while *C. yedoensis* ‘Somei-Yoshino’ enters this stage in mid-late April. There was little variation in the flower bud differentiation stage among different species and varieties. All of them initiate the early stage of flower bud differentiation in July, which persists for a considerable duration and ceases before the onset of dormancy. From late September to early October, the six species and varieties gradually enter the autumn discoloration period, which can persist for approximately 30 days. Similar to the flowering period, this period also holds high ornamental value.

Based on the results of growth index measurements and phenological period observations from 2016 to 2018, it can be concluded that all five species and varieties of *Cerasus* exhibited normal growth states and were capable of safely overwintering in Qingdao.

### 2.2. Study on Cold Tolerance of Cerasus Plants in Natural Cooling Process

According to the meteorological data of the test site, from 1 December 2016, to 18 February 2017, the temperature of the test site exhibited a trend of initially decreasing and then rising, reaching its lowest point of −8 °C around 22 January 2017, during the entire test sampling period. From 12 January 2017, to 25 January 2017, the average maximum temperature for half a month reached a minimum of 3.4 °C, and the average minimum temperature during this period also dropped to −3.8 °C. After 11 February 2017, the temperature began to rise slowly, but by 18 February 2017, the average minimum temperature had not risen above zero.

When subjected to low temperatures, the concentration of soluble proteins initially decreased and then subsequently increased (Figure 1a). The proline content fluctuated, but there was no obvious pattern (Figure 1b). The permeability of the cell membrane continuously decreased with decreasing temperature. (Figure 1c), leading to membrane lipid peroxidation and the accumulation of reactive oxygen species, resulting in higher levels of MDA, a product of lipid peroxidation (Figure 1d). The activity of SOD increased with decreasing temperatures but decreased with rising temperatures (Figure 1e). Furthermore, the activities of POD enzymes in plants exhibited an increasing trend as the temperature decreased (Figure 1f).

The average minimum temperature over a two-week period exhibited a negative correlation with relative conductivity, MDA content, SOD enzyme activity, and POD enzyme activity across most cherry species and varieties, while it showed a positive correlation with the free proline content in the five cherry species and varieties (Figure 2).

The membership function method was used to comprehensively assess the cold tolerance of the five *Cerasus* species and varieties. Based on the average membership value, a simple ranking was conducted to visually demonstrate their cold tolerance, as shown in Table 3. According to the average membership value of each variety, the cold tolerance ranking of the five *Cerasus* species and varieties is as follows: *C. pseudocerasus* ‘Introtsa’ > *C. discoidea* > *C. subhirtella* ‘Autumnalis’ > *C. dielsiana* > *C. campanulata* ‘Youkou’.

In this experiment, we employed quantitative analysis and principal component analysis to establish correlations. Subsequently, we utilized the membership function method to derive membership values. Finally, comprehensive evaluation, a commonly used method in assessing cold tolerance, was performed, as it has been widely applied in the identification of cold tolerance in grape (*Vitis vinifera* L.) [29], pear (*Pyrus* species) [30], apple (*M. pumila* Mill) [31], and other plants. This approach enables a comprehensive analysis of plant cold tolerance during natural overwintering.

### 2.3. Analysis of Nutrient and Hormone Content Changes during Flower Bud Differentiation of C. yedoensis ‘Somei-Yoshino’ and C. subhirtella ‘Autumnalis’

#### 2.3.1. Observation of Flower Bud Internal Morphology during Differentiation

The examination of the morphological differentiation of flower buds in the twice-a-year flowering variety *C. subhirtella* ‘Autumnalis’ and the control variety *C. yedoensis* ‘Somei-Yoshino’ showed that both cultivars displayed comparable stages of differentiation, consisting of seven stages. These stages included the undifferentiated stage, initial stage of differentiation, flower primordium differentiation stage, calyx primordium differentiation stage, petal primordium differentiation stage, stamen primordium differentiation stage, and pistil primordium differentiation stage (Figure 3).

#### 2.3.2. Changes in Soluble Sugar and Soluble Protein Contents in Leaves during Flower Bud Differentiation

There was an observable pattern in the soluble protein content in the leaves of *C. yedoensis* ‘Somei-Yoshino’ and *C. subhirtella* ‘Autumnalis’ (Figure 4a). It showed an overall trend of initial decrease followed by an increase. Both cherry blossom varieties experienced rapid consumption of soluble proteins during the morphological differentiation stage of the floral buds, with the lowest content observed from the floral primordium differentiation stage to the calyx primordium differentiation stage. Following this, the amount of soluble proteins in the leaves exhibited a gradual increase after the stage of petal primordium differentiation, followed by a declining trend until the stage of pistil primordium differentiation. Notably, while the soluble protein content in *C. yedoensis* ‘Somei-Yoshino’ decreased during the pistil primordium differentiation stage, *C. subhirtella* ‘Autumnalis’ maintained a consistently high level. Moreover, during flower bud differentiation, the leaves of both longer and shorter branches of *C. subhirtella* ‘Autumnalis’ exhibited higher soluble protein content.

The experimental findings revealed that during the flower bud differentiation period, there was an overall trend of initial increase followed by a decrease in the soluble sugar content in the leaves of *C. yedoensis* ‘Somei-Yoshino’ and *C. subhirtella* ‘Autumnalis’ (Figure 4b). The process of flower bud differentiation in plants requires the utilization of a significant amount of soluble sugars, and a sufficient supply of carbon assimilates promotes this process. Some soluble sugars serve as respiratory substrates, while others are converted into essential nutrients needed for flower bud differentiation [32]. It is worth mentioning that, in the flower bud differentiation process of *C. subhirtella* ‘Autumnalis’, the leaves on shorter branches showed notably higher levels of a soluble sugar content when compared to leaves on other branches. Before entering flower bud differentiation, the soluble protein content in the leaves of ‘October cherry’ and ‘Ranai Yoshino’ rapidly decreased, while the soluble sugar content increased during the same period.

#### 2.3.3. Changes in Hormone Content in Flower Buds during Flower Bud Differentiation

Both Flowering cherry varieties exhibited relatively high levels of ABA and GA3 (Gibberellin A3) during the physiological differentiation stage of the flower buds (Figure 5b). As the morphological differentiation stage began, the ABA content gradually increased, reaching a relatively high level upon the completion of some flower bud differentiation (Figure 5a). From the physiological differentiation stage to the flower primordium differentiation stage, the ABA and IAA contents in *C. subhirtella* ‘Autumnalis’ flower buds were significantly higher compared to *C. yedoensis* ‘Somei-Yoshino’. Morphological sectioning revealed a faster rate of flower bud differentiation in *C. subhirtella* ‘Autumnalis’ during these two periods [33]. The content of ZR in the flower buds reached its maximum level during the physiological differentiation stage (Figure 5d). It then gradually decreased during the morphological differentiation stage, along with a decrease in both ZR and IAA content. During the pistil primordium differentiation period, the ZR and IAA contents in *C. subhirtella* ‘Autumnalis’ flower buds showed an upward trend (Figure 5c). In contrast, *C. yedoensis* ‘Somei-Yoshino’ did not exhibit this change. Furthermore, the ZR and IAA contents in the flower buds of *C. subhirtella* ‘Autumnalis’ with shorter branches demonstrated a faster increase compared to those with longer branches. Phenology observations indicated a higher germination rate of flower buds in *C. subhirtella* ‘Autumnalis’ with shorter branches during autumn flowering.

## 3. Materials and Methods

### 3.1. Experimental Materials

In February 2016, nine cherry blossom species and varieties (Table 4) were introduced from Wuxi, Jiangsu Province, and planted in the seedling base of Qingdao Agricultural University in Jiangjia Tuzhai, Laoshan District. Eight representative medium-sized sample plants were selected for each type, totaling 40 plants, as the subjects of observation.

Nutrient and hormone level data during bud differentiation were obtained from the campus of Qingdao Agricultural University (120°39′ E, 36°32′ N). The cherry blossom varieties chosen for the experiment included the secondary flowering variety *C. subhirtella* ‘Autumnalis’, *C. subhirtella* ‘Accolade’, the annual flowering variety *C. yedoensis* ‘Somei-Yoshino’, and *C. serrulata* var. *lannesiana* ‘Matsumae Benihigoromo’ [34]. Three plants with similar growth vigor and size were selected as test materials for each variety.

Qingdao, located in Shandong Province, falls within the warm temperate monsoon climate zone, with an average annual temperature of 13 °C and average annual precipitation of 750 mm. The park’s soil mainly consists of medium-grade soil [35].

The meteorological data for the test site (Figure 6) are organized in Table 2. As depicted in Table 5, from 1 December 2016, to 18 February 2017, the ambient temperature at the test site followed a pattern of initially decreasing and then increasing. It reached its lowest point of −8 °C around 22 January 2017, during the entire sampling period. Between 12 January 2017, and 25 January 2017, the average maximum temperature for half a month hit a minimum of 3.4 °C, and the average minimum temperature also reached −3.8 °C during this period. After 11 February 2017, the temperature gradually started to rise, but by 18 February 2017, the average minimum temperature remained below freezing.

### 3.2. Experimental Methods

#### 3.2.1. Phenological Observation

Eight healthy and pest-free trees were selected for each observed tree species, with the locally suitable variety *C. yedoensis* ‘Somei-Yoshino’ serving as the control. The flower buds taken in the experiment were fixed with FAA (70% ethanol, 38 formaldehyde, acetic acid in a volume ratio of 90:5:5) and stored in a refrigerator at 4 °C. The flower buds were prepared by slicing them through the traditional paraffin wax method. A rotary microslicer (LEICARM2126RT) was used for slicing, and they were then stained with oxylin. Afterward, they were sealed with neutral gum, examined under a microscope, and photographed. We made daily observations during the flowering and leaf-spreading season, covering various stages, including bud sprouting, the start of new shoot growth, autumn leaf discoloration, as well as fruit and seed maturation and shedding. These observations were made every two days. Additionally, climatic conditions were recorded (Table 6).

#### 3.2.2. Growth Indicators

We measured the following growth parameters: ground diameter (diameter at a point 20 cm above the ground), plant height (distance from the base of the stem to the top of the main stem), crown width (average width in both the north–south and east–west directions), branch point height (height from the base of the stem to the meristem of the main stem), growth index, and the annual increase in diameter.
$$\mathrm{Growth}\;\mathrm{index}\;\mathrm{Gi}=\frac{\mathrm{Al}}{\mathrm{AL}}\;\mathrm{Growth}\;\mathrm{index}\;\mathrm{GiGrowth}\;\mathrm{index}$$

Al is the annual increment of diameter, AL = $\frac{\sum \mathrm{Al}}{N}$ is the average annual increment of diameter, N = Growth years.

#### 3.2.3. Physiological Index Determination

Using a mortar, a 0.5 g sample of plant branches was ground to a powder in an ice bath. Subsequently, 10 mL of 7.8 phosphoric acid buffer was added, and the homogenate was transferred to a 15 mL centrifuge tube. The tube was then centrifuged at 4 °C for 20 min at 4000 r/min using a high-speed refrigerated centrifuge. The resulting supernatant was collected in a test tube and stored in a refrigerator at 4 °C.

To measure relative conductivity, branches were directly cut into 2 mm segments after washing. These segments were immersed in 20 mL of distilled water for 18 h, and the conductivity (R) was measured. Subsequently, the tube containing the material was placed in a 100 °C water bath and boiled for 30 min, after which the conductivity (R0) was measured to determine the relative conductivity. The content of malondialdehyde (MDA) in plants was determined using the thiobarbituric acid method, following the experimental method of Li Hesheng [36]. Superoxide dismutase (SOD) activity in plants was assessed using the nitroblue tetrazolium (NBT) photochemical reduction technique. Peroxidase (POD) activity in plants was determined using the guaiacol method. The concentration of free proline in plants was quantified using the sulfosalicylic acid method.

In late June 2020, branches were collected from *C. subhirtella* ‘Autumnalis’ and *C. yedoensis* ‘Somei-Yoshino’ after flower bud differentiation. Samples were collected every 15 days for analysis. Three to five branches with healthy growth and no pests were randomly selected each time, and the leaves were washed and dried before measurement. The determination of soluble protein and soluble sugar followed the experimental techniques of plant physiology [37]. Absorbance was measured using a Hitachi UH5300 double beam spectrophotometer in the School of Landscape Architecture and Forestry laboratory at Qingdao Agricultural University.

The contents of zeatin riboside (ZR), GA3, IAA, and ABA in samples from different periods were determined using an Agilent 6460 Triple Quad LC/MS liquid chromatography mass spectrometry in the Central Laboratory of Qingdao Agricultural University. The extraction and purification of native hormones from the sample varieties were conducted as follows: 0.5 g of cherry blossom bud samples were precisely weighed and placed in a mortar. The samples were ground under the freezing conditions of liquid nitrogen, and extraction was performed twice under a temperature of 4 °C. The supernatant was combined, and the extract was purified using a C18 solid-phase extraction column. It was then filtered through a 0.2 µm organic microporous filter membrane for testing. The operations were conducted away from light and at a temperature below 4 °C. The instrument parameters were comprehensively referenced from the methods of Zhong et al. [38] and Pan et al.

## 4. Discussion

Through the assessment of growth indices and phenological periods, it was observed that *C. dielsiana*, *C. pseudocerasus* ‘Introtsa’, and *C. subhirtella* ‘Autumnalis’ exhibited better cold adaptation compared to *C. discoidea* and *C.*‘Youkou’. The flowering patterns of *C. subhirtella* ‘Autumnalis’, *C. pseudocerasus* ‘Introtsa’, *C. dielsiana*, and *C.* ‘Youkou’ occurred sequentially, with the second flowering period of *C. subhirtella* ‘Autumnalis’ lasting from late September to early November and exhibiting a longer duration of 54 days.

During the natural overwintering process, the relative conductivity of plants increased as the temperature decreased, consistent with the findings of Bai et al. [39] and Li et al. [40]. The accumulation of MDA content in plants increased as the temperature of their surroundings decreased, rendering them more susceptible to low-temperature damage [41]. When plants are exposed to low temperatures, they generate a significant number of superoxide anion free radicals, which possess potent oxidative capabilities and can be harmful to organisms. SOD is an enzyme present in plants that scavenges superoxide radicals, converting them into hydrogen peroxide and oxygen to reduce toxicity. Luo et al. [42] and Che et al. [43] similarly observed a negative relationship between SOD enzyme activity and temperature. When SOD breaks down superoxide anion radicals, it results in the accumulation of hydrogen peroxide in plants, which can lead to harmful oxidation. The experimental findings suggest that under particular conditions and within specific temperature ranges, lower temperatures are associated with increased activity of POD and CAT enzymes in plants. This enhanced enzyme activity helps plants better withstand cold-induced damage. These findings align with the research of Wang et al. [44] and Feng et al. [45].

When plants experience low-temperature damage, the content of soluble proteins in certain varieties initially decreases and then increases. This is because, when plants are initially exposed to low temperatures, membrane proteins dissociate from the membrane and become free proteins, increasing soluble protein content. As the temperature further decreases, the harmful effects of low temperature intensify, leading to the further breakdown of soluble proteins into amino acids, resulting in a downward trend [46]. Subsequently, as the temperature continues to decrease, the soluble protein content rebounds, facilitating water retention in cells and safeguarding the normal physiological activities of plants. This enables them to withstand low-temperature damage. These findings are consistent with the research conducted by Luo et al. [47].

The average minimum temperature during a two-week period demonstrates a negative correlation with relative conductivity, MDA content, SOD enzyme activity, and POD enzyme activity in the majority of cherry tree species and varieties. Additionally, it is positively correlated with the free proline content of these species and varieties. These findings align with the research of Ning et al. [48], and Pan et al. [49], but contradict the research of Li et al. [50]. The relationship between free proline content and cold tolerance remains unclear [51]. However, the role of proline in cold tolerance is highly complex, as it serves as an osmoregulation substance, protecting proteins from damage, and actively scavenging reactive oxygen species [52]. Further research can analyze the differences in free proline content between species and varieties during natural overwintering by utilizing more data.

Soluble proteins play a crucial role in plant physiology as signaling molecules and structural substances during the process of flower bud differentiation [53]. Research indicates that the accumulation of proteins is advantageous for flower bud differentiation, and a substantial quantity of protein is utilized during this process [54]. The soluble protein content in *C. subhirtella* ‘Autumnalis’ leaves increased after flower bud differentiation. However, annual flowering varieties, such as ‘Somei-Yoshino ‘, showed no significant increase in soluble protein content. This indicates that the accumulation of soluble protein content contributes to the sprout of cherry blossom buds.

As soluble sugars are important energy substances, sucrose and D-glucose may also be the initial signals regulating the transition from physiological differentiation to morphological differentiation [38,39]. During the critical period of flower bud differentiation, the rapid increase in soluble sugar content in leaves promotes this process [55,56]. Our experimental findings additionally revealed that the leaves on shorter branches of the secondary flowering varieties contained higher levels of soluble sugar compared to other varieties and longer branches. Simultaneously, the process of flower bud differentiation was completed earlier, and the overall plumpness was generally higher in these cases. This could potentially serve as the material foundation for autumn flower bud sprouting.

The experimental results are consistent with previous studies, suggesting that high levels of ABA and ZR can promote plant flower bud differentiation [57]. ABA exhibits a high content before flower bud differentiation, contributing to the initiation of morphological differentiation, while the increase in GA3 content during physiological differentiation supports the progress of physiological differentiation in cherry flower buds [58,59]. The high content of ABA and GA3 aids in the physiological differentiation of cherry flower buds, which aligns with research on changes in endogenous hormone content during flower bud differentiation in *Ficus carica* Linn. [60] and Bougainvillea Comm. ex Juss [61]. Notably, the higher content of ABA and IAA in the flower buds of *C. subhirtella* ‘Autumnalis’ accelerates the differentiation rate during the early and flower primordia differentiation stages compared to *C. yedoensis* ‘Somei-Yoshino’, creating favorable conditions for *C. subhirtella* ‘Autumnalis’ to bloom before entering ecological dormancy in autumn. On the other hand, a reduced amount of ZR and IAA contributes to the process of morphological differentiation in cherry flowers, as observed in *Olea europaea* L. and *Osmanthus* sp. [59,62,63]. Additionally, a higher ZR content promotes the physiological differentiation of plant flower buds. Based on these findings, it is speculated that the increase in ZR and IAA content facilitates the completion of flower bud differentiation in cherry blossoms and promotes the flowering of *C. subhirtella* ‘Autumnalis’ before external environmental conditions induce its ecological dormancy.

## 5. Conclusions

In summary, we conducted observations and measurements of phenology, growth indicators, and physiological indicators in five cherry blossom species. Based on our findings, the ranking of their cold tolerance is as follows: *C. pseudocerasus* ‘Introtsa’ > *C. discoidea* > *C. subhirtella* ‘Autumnalis’ > *C. dielsiana* > *C.* ‘Youkou’ cherry. The correlation analysis between physiological indicators and the average monthly minimum temperature revealed negative or significantly negative associations with relative conductivity, SOD activity, POD activity, MDA content, and soluble protein content. Additionally, we conducted a comparison of the morphological and physiological indicators between the once-a-year flowering variety *C. yedoensis* ‘Somei-Yoshino’ and the twice-a-year flowering variety *C. subhirtella* ‘Autumnalis’ during the flower bud differentiation process. Based on our findings, it is suggested that the higher soluble sugar content in the leaves of *C. subhirtella* ‘Autumnalis’ may contribute to the earlier completion and increased plumpness of flower bud differentiation in shorter branches, facilitating autumn flowering. The higher content of ABA and IAA in the flower buds of *C. subhirtella* ‘Autumnalis’ facilitates a faster differentiation rate during the early and flower primordia differentiation stages compared to *C. yedoensis* ‘Somei-Yoshino’, allowing *C. subhirtella* ‘Autumnalis’ to bloom before entering ecological dormancy in autumn. The flower bud differentiation process in cherries is complex, and the dynamic equilibrium of various hormones plays a crucial role in flower bud differentiation and secondary flowering.

## Figures and Tables

**Figure 1 plants-12-03980-f001:**
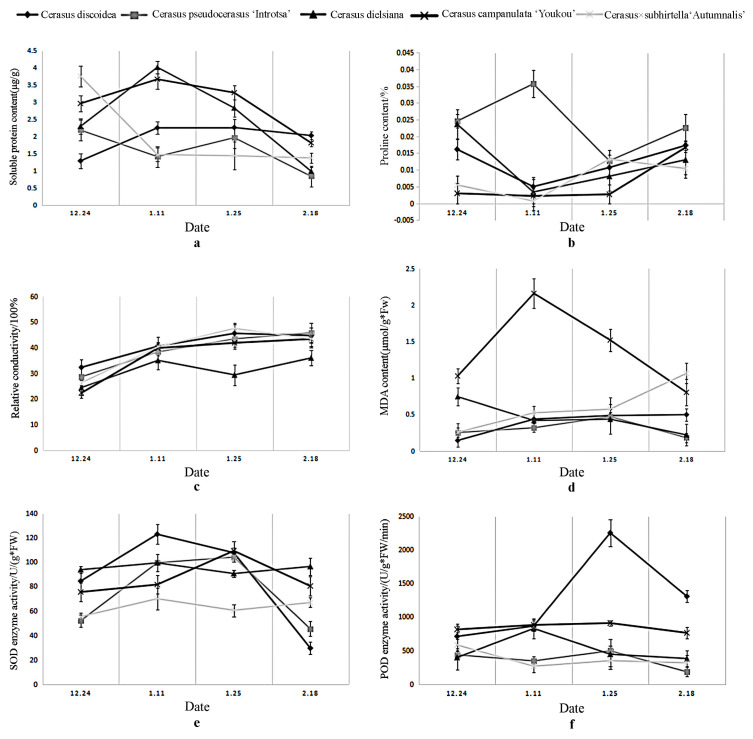
Changes in the content of various physiological indicators. (**a**): Changes in soluble protein content, (**b**): Change in proline content, (**c**): changes in relative conductivity, (**d**): Changes in MDA content, (**e**): Changes in SOD enzyme activity, (**f**): Changes in POD enzyme activity.

**Figure 2 plants-12-03980-f002:**
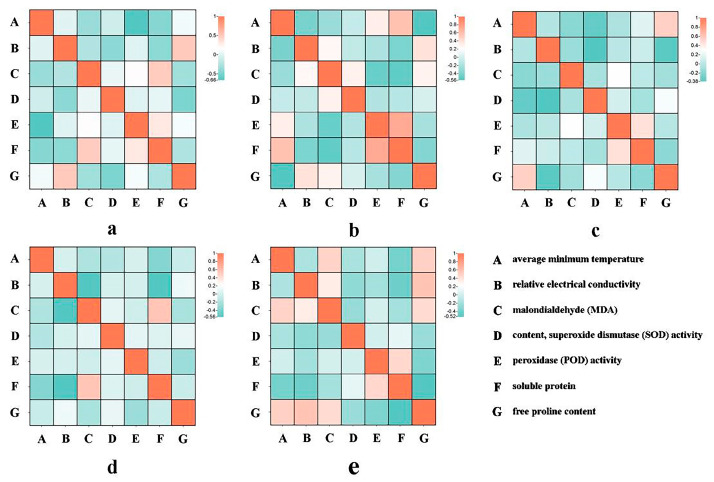
Correlation coefficient analysis of the average minimum temperature and the physiological indexes of five cherry blossoms (**a**): *C. discoidea*, (**b**): *C. pseudocerasus* ‘Introrsa’, (**c**): *C. subhirtella* ‘Autumnalis’, (**d**): *C. dielsianae*, (**e**): *C.* ‘Youkou’.

**Figure 3 plants-12-03980-f003:**
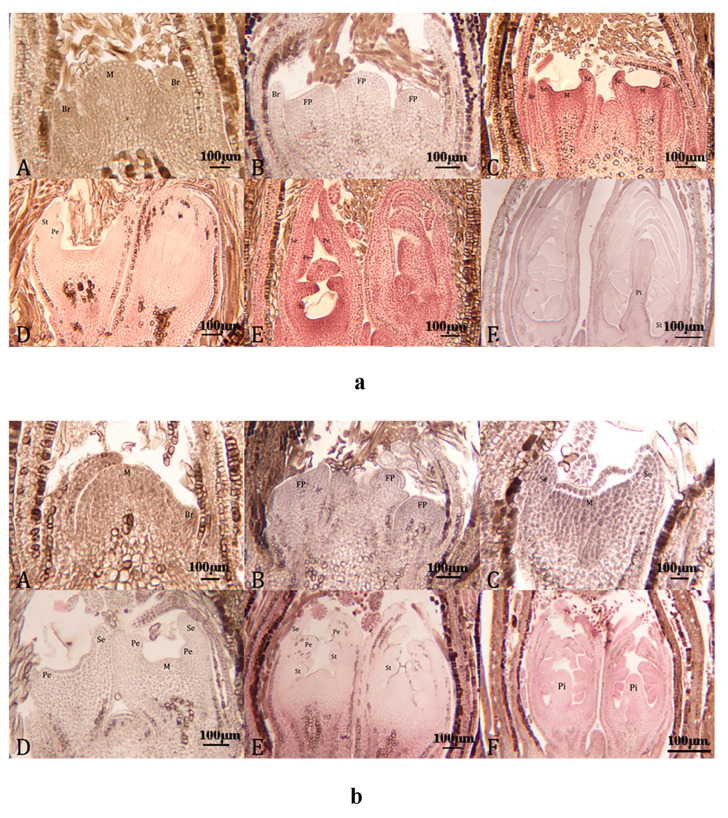
Images showing the morphological differentiation of *C. subhirtella* ‘Autumnalis’ and *C. yedoensis* ‘Somei-Yoshino’ flower buds in different periods (**a**): *C. subhirtella* ‘Autumnalis’; (**b**): *C. yedoensis* ‘Somei-Yoshino’). (**A**): Early differentiation (bract primordium differentiation); (**B**): Flower primordium differentiation; (**C**): Calyx primordium differentiation; (**D**): Petal primordium differentiation; (**E**): Stamen primordium differentiation; (**F**): Pistil primordium differentiation. M: Growth point; Br: Bract stem; FP: Flower bud primordia; Se: Sepal primordium; St: Stamen primordium; Pe: Petal primordia; Pi: Pistil primordia.

**Figure 4 plants-12-03980-f004:**
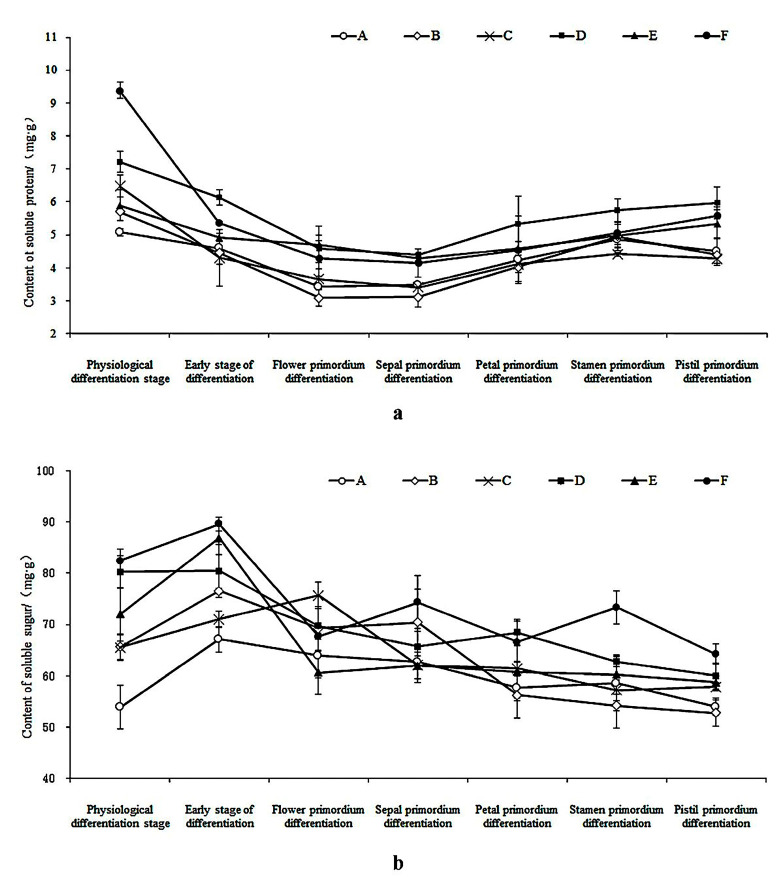
Changes in nutrient content in leaves of *C. yedoensis* ‘Somei-Yoshino’ and *C. subhirtella* ‘Autumnalis’ (**a**): soluble protein content; (**b**): soluble sugar content. A: Long branches front leaves of *C. yedoensis* ‘Somei-Yoshino’; B: Long branches posterior leaves of *C. yedoensis* ‘Somei-Yoshino’; C: Short branches leaves of *C. yedoensis* ‘Somei-Yoshino’; D: Long branches front leaves of *C. subhirtella* ‘Autumnalis’; E: Long branches posterior leaves of *C. subhirtella* ‘Autumnalis’; F: Short branches leaves of *C. subhirtella* ‘Autumnalis’.

**Figure 5 plants-12-03980-f005:**
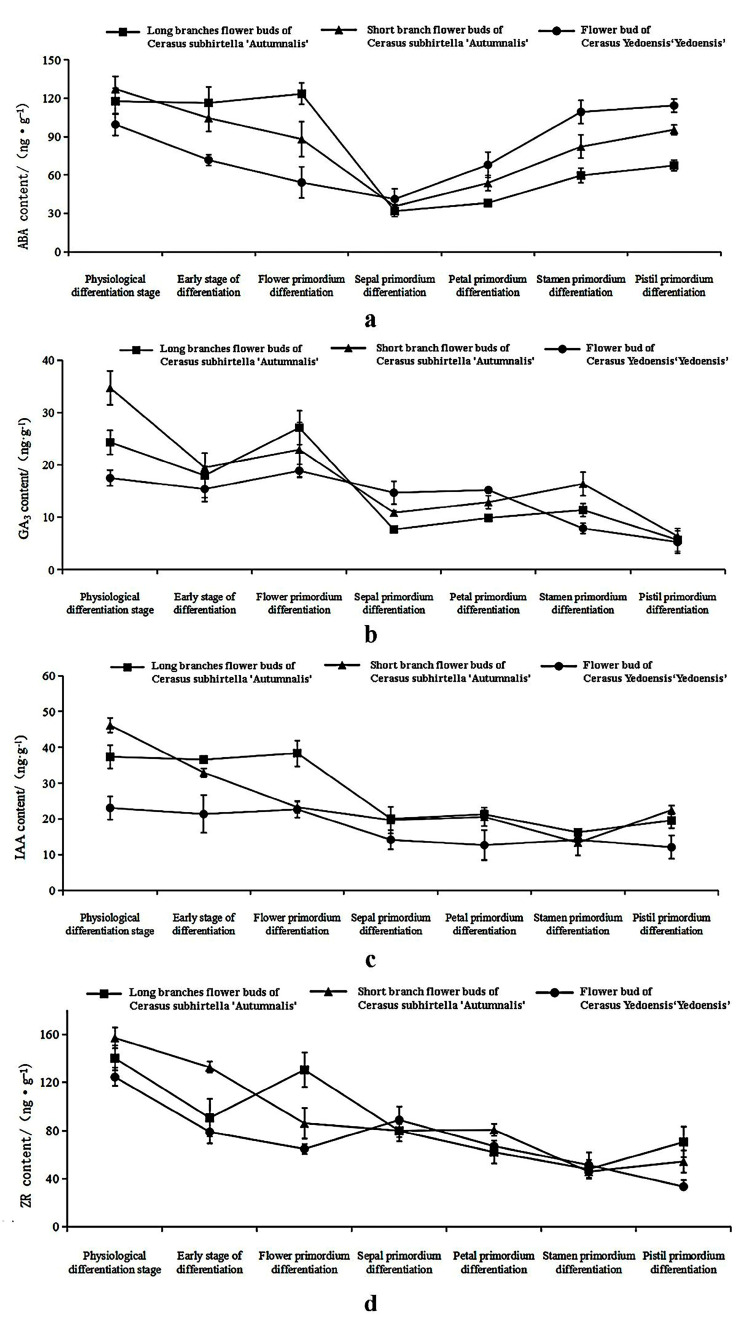
Dynamic changes in hormone content in flower bud differentiation (**a**): ABA; (**b**): GA3; (**c**): IAA; (**d**): ZR.

**Figure 6 plants-12-03980-f006:**
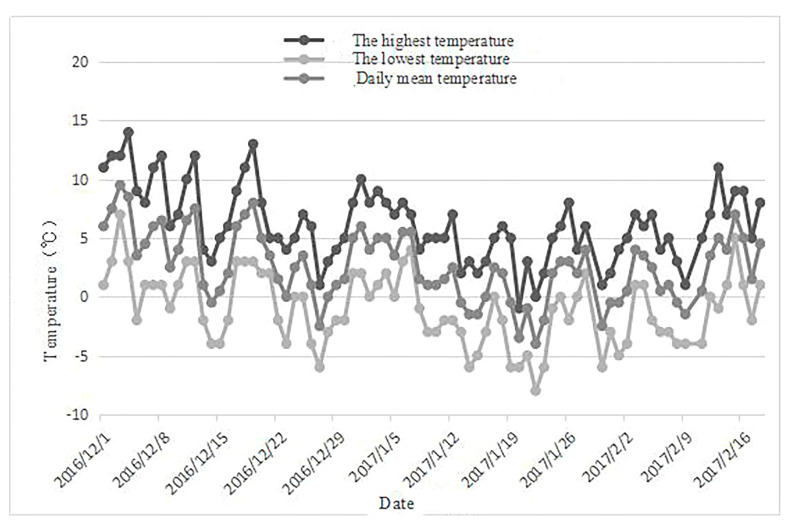
Trend chart of temperature change in experimental field.

**Table 1 plants-12-03980-t001:** Changes in growth indicators of cherry blossoms from 2016 to 2018.

Variety Name	Survey Year	Average Ground Diameter (cm)	Average Plant Height (cm)	Average Crown Width (cm)	Annual Growth of Average Ground Diameter (cm)	Growth Index
*Cerasus discoidea*	March 2016	3.98	313	139	*---------*	*---------*
	March 2017	4.08	320	144	0.10	0.77
	March 2018	4.24	325	148	0.16	1.23
*Cerasus pseudocerasus* ‘Introtsa’	March 2016	3.90	248	149	*---------*	*---------*
	March 2017	4.20	259	191	0.30	0.46
	March 2018	5.20	278	226	1.00	1.54
*Cerasus subhirtella* ‘Autumnalis’	March 2016	3.84	221	159	*---------*	*---------*
	March 2017	4.13	235	191	0.29	0.59
	March 2018	4.82	256	214	0.69	1.41
*Cerasus dielsiana*	March 2016	3.85	295	175	*---------*	*---------*
	March 2017	4.13	299	184	0.28	0.67
	March 2018	4.68	312	193	0.55	1.33
*Cerasus* ‘Youkou’	March 2016	3.51	227	130	*---------*	*---------*
	March 2017	3.74	234	138	0.23	0.78
	March 2018	4.10	248	147	0.36	1.22

**Table 2 plants-12-03980-t002:** Phenophase Observations for 2016–2018.

Variety Name	Years	Bud Enlargement	Flowering Period	Leaf Stage	Fruit Growth and Fruit Drop	Shoot Growth Period	Flower Bud Differentiation	Autumn Leaf Discoloration	Deciduous Period	Dormant Period
*Cerasus discoidea*	2016	End of February–03/09	03/10–04/11	04/05–04/20	04/20–05/15	04/09–09/15	Early 2007–mid-November	09/25–10/22	10/21–Early mid-November	Mid-November–mid-February
	2017	End of February–03/06	03/07–04/10	04/03–04/18	04/18–05/12	04/05–09/10	Early 2007–mid-November	09/20–10/20	10/18–Early mid-November	Mid-November–mid-February
*Cerasus pseudocerasus* ‘Introtsa’	2016	Early March–03/16	03/17–04/03	04/02–04/18	04/22–05/18	04/09–09/16	Mid-July–mid-November	10/01–10/30	11/07–Late November	Late November–early March
	2017	Early March–03/14	03/15–04/03	04/02–04/15	04/20–05/15	04/07–09/15	Mid-July–mid-November	09/30–10/30	11/05–Late November	Late November–early March
*Cerasus subhirtella* ‘Autumnalis’	2016	Early March–03/14 Mid-September– 09/20	03/15–04/15 09/21–11/11	04/12–04/19	04/23–05/10	04/13–09/22	Early July–mid-November	09/30–11/07	10/28–Late November	Late November–late February
	2017	Early March–03/13 Mid-September– 09/19	03/13–04/12 09/20–11/10	04/10–04/17	04/20–05/07	04/12–09/20	Early July–mid-November	09/28–11/05	10/28–Late November	Late November–late February
*Cerasus dielsiana*	2016	Early March–03/18	03/19–04/09	04/07–04/22	04/20–05/22	04/09–09/20	Mid-July–mid-November	09/27–11/04	11/09–Mid-November	Mid-November–early March
	2017	Early March –03/15	03/16–04/07	04/05–04/20	04/18–05/20	04/07–09/18	Mid-July–mid-November	09/25–11/02	11/08–Mid-November	Mid-November–early March
*Cerasus* ‘Youkou’	2016	Early March–03/18	03/19–04/18	04/14–04/22	04/25–05/30	04/13–09/26	Mid-July–mid-November	09/30–11/05	November/December–Late November	Late November–early March
	2017	Early March–03/17	03/18–04/15	04/12–04/21	04/23–05/30	04/12–09/25	Mid-July–mid-November	09/27–11/03	November/October–Late November	Late November–early March
*Carasus × yedoensis* ‘Somei-Yoshino’	2016	Mid-March–04/06	04/07–04/27	04/21–05/03	05/05–06/10	04/23–10/16	Mid-07–mid-to-late November	10/10–11/15	11/22–Late November	Late November–early April
	2017	Mid-March–04/04	04/05–04/25	04/19–05/01	05/03–06/08	04/21–09/24	Mid-07–mid-to-late November	10/08–11/14	11/20–Late November	Late November–early April

**Table 3 plants-12-03980-t003:** Membership function and cold tolerance ranking of physiological indexes in branches.

Species and Varieties	Relative Conductivity of Twigs	SOD Activity	POD Activity	MDA Content	Soluble Protein Content	Free Proline Content	Average Membership Function	Cold Tolerance Ranking
*discoidea*	0.658	0.512	0.655	0.853	0.630	0.254	0.594	②
*C. pseudocerasus* ‘Introtsa’	0.314	0.589	0.871	0.883	0.347	0.606	0.602	①
*C. subhirtella* ‘Autumnalis’	0.792	0.320	0.126	0.784	0.407	0.773	0.534	③
*C*. *dielsiana*	0.716	0.498	0.841	0.194	0.495	0.391	0.522	④
*C. ‘*Youkou’	0.323	0.545	0.303	0.544	0.406	0.839	0.493	⑤

**Table 4 plants-12-03980-t004:** Introduction of cherry.

Number	Plant	Ornamental Traits	Flowering Characteristics	Provenance
1	*C. subhirtella* ‘Autumnalis’	Semidouble light pink flower, blooming twice a year	It opens twice in spring and autumn, from mid to late March in spring, and from mid to late October until the end of November in autumn	*C. spachiana* × *Cerasus incisa* × Other cherry blossom varieties
2	*C. subhirtella* ‘Accolade’	Semidouble pink flower, blooming twice a year	It opens twice in spring and autumn, with the flowering period occurring from October to December each year, and from March to April the following year	*Cerasus sargentii* Rehd. × *C. spachiana*
3	*C. yedoensis’* Somei-Yoshino’	Main cultivated varieties in garden, single petal flower, pink white	It opens in late March in spring, with flowers appearing first and leaves following later	*C. speciosa* × *C. spachiana*
4	*Cerasus serrulata* var. *lannesiana* ‘Matsumae-Benihigoromo’	Double petal, pink, flowering in early April	Late March to early April in spring	*C. nipponica* × *C. lannesiana*
5	*C. pseudocerasus* ‘Introtsa’	Single petal flower, 5 petals, pink—Pink white	Mid to late March in spring	*Prunus campanulata × Prurus pseudo-cerasus*
6	*C.* ‘Youkou’	Single petal flower, 5 petals, light red	From late March to early April in spring, flowers bloom first and leaves follow later	*Cerasus* × *yedoensis*‘Amagi-yoshino’ × *P. campanulata*
7	*C. discoidea*	Single petal, 5 petals, pink	Spring blooms in March, with flowers appearing first and leaves following later	Chinese original species
8	*C. dielsiana*	Single petal, 5 petals, white or pink	From March to April in spring, the first leaves open or are near opening	Chinese original species
9	*C. spachiana f.* *ascendens*	Single white flower	Early March to early April in spring	Native to Japan

**Table 5 plants-12-03980-t005:** The average maximum temperature of the collection period, the lowest average temperature, the lowest temperature.

Sample Collection Period	Semi-Monthly Mean Maximum Temperature (°C)	Monthly Mean Minimum Temperature (°C)	Minimum Temperature During Collection Period (°C)
24 December 2016	7.1	0.1	−4
11 January 2017	6.1	−0.7	−6
25 January 2017	3.4	−3.8	−8
18 February 2017	5.6	−1.4	−6

**Table 6 plants-12-03980-t006:** Phenological observation standard.

Main Observation Period	Observing Standards
Bud expansion stage	Bud scales begin to separate, and young leaves or flower buds of fresh color appear at the top of the bud
Flowering stage	A little green appears at the tip of the flower bud, and approximately 80% of the flowers on the entire tree typically shed petals
Leaf spreading period	The first batch of leaflets, with 1–2 leaflets, spread out to half of the branches, eventually spreading out completely
Fruit growth and development and fruit drop period	The ovary begins to swell until more than half of the fruit falls off
New shoot growth period	It starts from the sprouting of leaf buds until the branches stop growing
Flower bud differentiation stage	The flower bud differentiation stage is observed through paraffin sectioning or the bud stripping method
Leaf color change period in autumn	Approximately 5% of the leaves of the entire plant begin to show autumn colors, with more than half of the leaves displaying autumn hues
Deciduous period	Approximately 5% of the leaves on the entire tree begin to fall off, with about 90% to 95% of the leaves on the entire tree eventually falling off
Dormancy period	All leaves fall off until the sap flows in the next year

## Data Availability

All detailed raw data can be founded in the Appendix A.

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
