# Peer review of "The Exploration of Flowering Mechanisms in Cherry Plants"

_plants, 2023, doi:10.3390/plants12233980_

Round 1

Reviewer 1 Report (Previous Reviewer 1)

Comments and Suggestions for Authors

Thank you for the detailed response to the questions and remarks.

The revision increased the quality of this manuscript to a certain level. Some improvements are still possible; other important remarks/comments are still ignored (e.g. headlines incomplete (e.g. type of organ); captions; quality of the figure 1). The grading "Average" is representing the maximum of the assessment combined with "minor revision". A further reviewer may be considered before acceptance and publication, especially to maintain the quality of this journal.

Comments on the Quality of English Language

 Minor editing of English language required.

Author Response

For research article

Response to Reviewer 1 Comments

1. Summary

2. Questions for General Evaluation

Reviewer’s Evaluation

Response and Revisions

Does the introduction provide sufficient background and include all relevant references?

Yes/Can be improved/Must be improved/Not applicable

Yes

Are all the cited references relevant to the research?

Yes/Can be improved/Must be improved/Not applicable

Yes

Is the research design appropriate?

Yes/Can be improved/Must be improved/Not applicable

Yes

Are the methods adequately described?

Yes/Can be improved/Must be improved/Not applicable

Yes

Are the results clearly presented?

Yes/Can be improved/Must be improved/Not applicable

Yes

Are the conclusions supported by the results?

Yes/Can be improved/Must be improved/Not applicable

Yes

3. Point-by-point response to Comments and Suggestions for Authors

Comments 1: The revision increased the quality of this manuscript to a certain level. Some improvements are still possible; other important remarks/comments are still ignored (e.g. headlines incomplete (e.g. type of organ); captions; quality of the figure 1). The grading "Average" is representing the maximum of the assessment combined with "minor revision". A further reviewer may be considered before acceptance and publication, especially to maintain the quality of this journal.

Response 1: Thank you for pointing that out. We agree with this opinion. Therefore, we have made supplementary changes to the subtitle of the article and explained to you the selection of plant materials (such as plant organs). We have added the environmental data of the plant growing place and replaced the figure 1 you mentioned. We have made changes to all of your questions in the original text and answered your questions individually in the review PDF.

4. Response to Comments on the Quality of English Language

Point 1:Minor editing of English language required.

Response 1: We have made a comprehensive revision of our language problems.

5. Additional clarifications

Reviewer 2 Report (New Reviewer)

Comments and Suggestions for Authors

This article reports the physiological and histological analyses and phenological observations of the blooming stages in flowering cherry species. The blooming phenology was characterized respectively in nine species of flowering cherry. Moreover, the changes in physiological indicators, such as MDA content, SOD and POD enzyme activities, soluble proteins and sugar contents, and phytohormones (ABA, GA3, IAA and Zeatin), were investigated in the floral buds during the dormancy to bud break stages. The histological observation of floral buds was also performed. This article concluded the ranking of cold tolerance among the flowering cherry species investigated. The correlations among the physiological and morphological indicators during the bud dormancy to blooming were characterized. This paper provides fundamental information related to flower blooming in cherry species with distinguishing characteristics in the blooming time.

Unfortunately, meteorological and environmental data such as temperature change and rainfall (or groundwater, or water status of each tree) are not provided in this manuscript. The environmental data of the tree-grown places is essential to discuss the physiological and phenological changes during the dormancy to blooming stage. The effects of temperature (low-temperature accumulation(chilling)/warm temperature) on the endo-/eco-dormancy breaks and the developmental speed and bud-break acceleration of the floral buds may be important in the discussion. Table 4 provides calendar data of the plant samples used in this study. However, we can’t know the environmental data (temperature, soil, and water) of the growing place (Qingdao Agricultural University in Jiangjia Tushai). Probably, many readers may have questions about the environmental condition of the grown place and its effects in the discussion of this article.

There are numerous errors in typing the cultivar name of ‘Somei-Yoshino’ throughout the manuscript.

‘Someo-Yoshino’ >> ‘Somei-Yoshino’

In Table1 and L.120; ‘Mastumae Benihigoromei’ >> ‘Mastumae Benihigoromo’

In Table 1, Table 3, Table 4, and Table 5; C. campanulata ‘Youkou’ >> ‘Youkou’ is a hybrid between C x yedoensis and P.(C.) campanulata, therefore the scientific name of ‘Youkou’ is not C. campanulate.

In Table 4; Cerasus Yedoensis ‘Yedoensis’ >> Carasus x yedoensis ‘Somei-Yoshino’

Round 2

Reviewer 2 Report (New Reviewer)

Comments and Suggestions for Authors

The manuscript has been properly improved by adding meteorological data. I did not notice minor errors furthermore.

Comments on the Quality of English Language

It's OK, I could understand your English text.

Author Response

This manuscript is a resubmission of an earlier submission. The following is a list of the peer review reports and author responses from that submission.

Round 1

Reviewer 1 Report

Comments and Suggestions for Authors

Briefly, the manuscript “The Exploration of Flowering Mechanisms in Cherry Plants” is very ambitious in the approach. However, it contains many deficiencies in all sections/chapters. “Introduction”, no reading flow; the analytical area is geared to buds as well as to leaves and branches, which, however, is not brought into harmony in a systematic and goal-oriented manner. The temperatures often mentioned are not shown in the manuscript. Statements made in this context cannot be understood; certainly not to the cold “resistance” or cold “tolerance”, which differ in definition. The content of Table 2 is not based on international standards; Table 3 – also - no statistical analysis; Table 4 data cannot be traced. The requirements for publication of the journal “Plants” are not fulfilled. All comments/remarks, see file.

Comments on the Quality of English Language

English must be checked by a native speaker/English service.

Reviewer 2 Report

Comments and Suggestions for Authors

In the figure 3 ,  refered as "a: C. subhirtella 'Autumnalis'; b: C. yedoensis 'Someo-yoshino", the figure is not of different species. a and b are the same photos, only with different contrast. This figure must be replaced.

It is also important, in the material and methods, to describe the histological techniques used, at least the  inclusion product and the dye used.

Reviewer 3 Report

Comments and Suggestions for Authors

The paper untitled ‘The Exploration of flowering mechanisms in cherry plants’ investigated several aspect of plant physiology of cherry trees during the flowering period and compared different species and varieties. The state of the art is clear but the objectives and methodology need clarifications and the results are not always well described and there are no statistical analyses. The discussion section needs also improvement. Overall, the paper needs strong improvement before publication

The title is quite broad and deserves to be more precise regarding the main objectives and results of the paper

Introduction:

Line 32: Please put ‘Cerasus’ in italics.

Line 74: Please give the complete name of MDA, POD, SOD at first mention.

Lines 99-101: Please add references for the sentence.

Lines 103-109: You describe what is done during your experiment but the objectives of the study are not clearly stated. Please clarify the aim of the study and the research questions.

Material and methods

Line 113, you mention 5 species and varieties but there are 9 species/varieties described in Table 1. Please clarify.

Lines 113-116: You mention introduction of species from one region of China to another but you do not precise which differences are between these regions. Please precise the importance of this information regarding the aim of the paper and give more information regarding the climate conditions of the mentioned regions for the readers that are not familiar with the geography and climate of China.

Line 118: The exact sampling material and time (which organ, which growing stage etc…) for nutrient and hormonal analysis in not clear. Please precise.

Line 126: ‘The park’s soil mainly consists of medium …..’ please complete the sentence.

Line 134: The reference to Table 2 must come before as it has nothing to do with the climatic conditions. The climatic conditions are not described.

Line 142 ‘Growth index GiGRowth index…. this is not clear

Line 145: What do you mean by ‘physiological index determination’?

Lines 146-150: The extraction with phosphoric acid was done to quantify which compound? Please clarify

Lines 157-160: please give more precision about the protocols of SOD, POD and proline quantification or give references to articles describing the methods.

Line 165-166: What do you mean by ‘techniques of plant physiology’?

Line 170-180: Please be clearer about the samples where the hormones were quantified. It is mentioned ‘from different period’, ‘the samples varieties’, ‘bud samples’….. it is note really clear.

From the material and methods, it is neither clear which measurements was performed on which species/varieties

There is no information about the statistical analyses performed in the paper

Results

Line 183: What do you mean by ‘the growth conditions was favorable’? It is not clear and precise.

Table 3 gives only means, please add standard deviations or other indicators of variability and statistical comparisons

Line 197: When you mention that the phenological period was earlier. What do you mean by ‘phenological period’? Which one? Do you mean the flowering period?

Line 219: You mention cell membrane permeability and electrolyte leakage but these techniques were not explained in the material and methods.

Line 219-251: The description of the biochemical compounds in the result section is really short and there is no information about differences between dates or between species/varieties and there are no statistical comparisons. I do not see the link between the described results and the cold tolerance mentioned in the title as we have no information about temperature or cold treatment.

Lines 238-244: You mention principal component analysis, correlation graphs, comprehensive evaluation…. All these statistical methods must be described in the material and methods section. There is not enough information to understand exactly what was done and all results are not presented.

The Figure 1 caption needs to be completed with more information so that the figure could be self-understandable.

Figure 3. Please explain the abbreviation used in the figures in the figure caption

Lines 269-28 Again there are no statistical analyses

Figure 4: What do you mean by physiological differentiation stage? Please add the meaning of A, B, C, D, E, F in the figure caption.

Lines 288-291: Some information of discussion are in the Results section and not in the Discussion. Please do clearly either a ‘results and discussion’ section or separate ‘results’ and ‘discussion’ in two separate sections but not a mix of both.

Figure 5 and lines 301-3019: Again statistical comparisons are missing.

Discussion

The results are over-interpreted in the discussion. For example line 330, the results obtained do not make it possible to directly link the modifications of MDA contents to the susceptibility to temperature. Several parameters could explain the observed results. Much of the discussion revolves around resistance to low temperatures but the temperatures to which the plants were exposed are not even presented in the article.  Lines 370-380, there is also an overinterpretation of the results because the results presented are not sufficient to support the proposed hypothesis.

Comments on the Quality of English Language

As far as I can tell the English is clear